# Ethnic differences in current smoking and former smoking in the Netherlands and the contribution of socioeconomic factors: a cross-sectional analysis of the HELIUS study

Rachel Brathwaite,[1] Liam Smeeth,[1] Juliet Addo,[1] Anton E Kunst,[2] Ron J G Peters,[3] Marieke B Snijder,[2] Eske M Derks,[4,5] Charles Agyemang[2]

[1]Department of Non-Communicable Disease Epidemiology, London School of Hygiene and Tropical Medicine, London, UK
[2]Department of Public Health, Academic Medical Center, University of Amsterdam, Amsterdam, The Netherlands
[3]Department of Cardiology, Academic Medical Center, University of Amsterdam, Amsterdam, The Netherlands
[4]Department of Psychiatry, Academic Medical Center, Amsterdam, The Netherlands
[5]QIMR Berghofer, Translational Neurogenomics group, Brisbane, Australia

**Correspondence to**
Rachel Brathwaite; Rachel.Brathwaite@lshtm.ac.uk

## ABSTRACT

**Objectives** Data exploring how much of the ethnic differences in smoking prevalence and former smoking are explained by socioeconomic status (SES) are lacking. We therefore assessed ethnic differences in smoking prevalence and former smoking and the contribution of both educational level and occupational-related SES to the observed ethnic differences in smoking behaviour.

**Methods** Data of 22 929 participants (aged 18–70 years) from the multiethnic cross-sectional Healthy Life in an Urban Setting study in the Netherlands were analysed. Poisson regression models with a robust variance were used to estimate prevalence ratios.

**Results** Compared with the Dutch, after adjustment for age and marital status, smoking prevalence was higher in men of Turkish (prevalence ratio 1.69, 95% CI 1.54 to 1.86), African Surinamese (1.55, 95% CI 1.41 to 1.69) and South-Asian Surinamese origin (1.53, 95% CI 1.40 to 1.68), whereas among women, smoking prevalence was higher in Turkish, similar in African Surinamese but lower in all other ethnic origin groups. All ethnic minority groups, except Ghanaians, had a significantly lower smoking cessation prevalence than the Dutch. Socioeconomic gradients in smoking (higher prevalence among those lower educated and with lower level employment) were observed in all groups except Ghanaian women (a higher prevalence was observed in the higher educated). Ethnic differences in smoking prevalence and former smoking are largely, but not completely, explained by socioeconomic factors.

**Conclusions** Our findings imply that antismoking policies designed to target smoking within the lower socioeconomic groups of ethnic minority populations may substantially reduce ethnic inequalities in smoking particularly among men and that certain groups may benefit from targeted smoking cessation interventions.

## INTRODUCTION

Ethnic differences in mortality rates and prevalence of lifestyle risk factors for diseases have been observed in the Netherlands.[1] Smoking is a leading lifestyle risk factor that accounts

### Strengths and limitations of this study

► The Healthy Life in an Urban Setting study used very large sample sizes in each ethnic group.
► Participants of the different ethnic groups were recruited in a systematic manner to reflect a representative sample of adults in each ethnic group living in Amsterdam.
► The same methodology was used to recruit participants from the different groups.
► Smoking status may be under-reported since biochemical methods were not used to verify self-reported smoking statuses.
► Smoking cessation was assessed by using quit ratios, which is a cumulative measure of a number of former smokers in the ethnic group, which occurred over time and not necessarily an indicator of a recent smoking cessation or assessment at the individual level.

for premature morbidity and mortality from several diseases in populations worldwide.[2] Higher rates of smoking, as well as lower rates of former smoking are more present in disadvantaged than advantaged groups in society.[3–5] Most ethnic minority groups may be of lower socioeconomic status (SES) than European host populations.[6] Evidence suggests variations in smoking prevalence across ethnic groups in high-income countries.[7 8] Differences in smoking prevalence among ethnic minority groups in European countries may be a reflection of their premigratory smoking status, or factors of the postmigratory environment, which influences smoking behaviour.[9 10] Smoking is associated with socioeconomic,[11] social and cultural factors, which undergo changes following migration to new environments. In the Netherlands, researchers observed socioeconomic

**BMJ**

gradients in smoking (smoking prevalence is higher as you have a lower socioeconomic group) among some ethnic minority populations, but not in sub-Saharan ethnic groups.[11] Inverse socioeconomic gradients in smoking cessation (smoking cessation rate is lower as you have a lower socioeconomic group) were recognised in other research.[12] Lower SES is associated with higher smoking possibly through mechanisms that include using smoking as a coping mechanism for stressful lives,[13 14] and less access to more costly and more effective options for smoking cessation.[15 16]

Few studies examined how much these differences in smoking and former smoking among ethnic minority groups relative to the Dutch were explained by socioeconomic factors using both educational level and occupational level as socioeconomic indicators.[11 17]

In addition to ethnic differences in smoking, there are gender differences in tobacco usage, smoking behaviour,[18] acceptability of smoking, self-reporting of smoking, smoking cessation and factors that may influence the continuation of smoking globally.[19] It is well documented that men have higher smoking rates and higher daily cigarette consumption compared with women, but this gender gap may vary among different ethnic groups and populations. We aimed to assess ethnic differences in smoking prevalence and former smoking among the Dutch, Ghanaian, Moroccan, Turkish, African Surinamese and South-Asian Surinamese origin ethnic groups in the Netherlands, and to study to what extent these ethnic differences could be possibly explained by differences in SES. In order to be able to address this general aim, we checked whether smoking prevalence and former smoking was related to low SES among those of Dutch origin and among the ethnic minority groups. We also calculated age-standardised prevalences of current smoking and former smoking by ethnic group and gender, and compared differences in smoking prevalence and former smoking in each minority ethnic group relative to the Dutch host population.

## METHODS
### Study design and setting
Baseline data from the multiethnic Healthy Life in an Urban Setting (HELIUS) study were used. Ethical approval was granted by the Institutional Review Board of the Academic Medical Center Amsterdam, and all participants provided written informed consent. In brief, HELIUS is investigating the patterns of health and healthcare utilisation among the largest ethnic groups resident in Amsterdam, the Netherlands, including the Dutch host population and the ethnic minority groups of Ghanaian, Moroccan, Turkish, South-Asian Surinamese and African Surinamese origin.[20]

### Participants
Participants aged 18–70 years were randomly sampled from the Amsterdam municipal population register,

stratified by ethnicity. Participants' ethnicity was defined according to the country of birth of the participant as well as that of his/her parents, which is currently the most widely accepted and most valid assessment of ethnicity in the Netherlands.[21] Specifically, a participant was considered as of ethnic minority origin if he/she fulfilled either of the following criteria: 1) he/she was born abroad and has at least one parent born abroad (first generation) or 2) he/she was born in the Netherlands but both his/her parents were born abroad (second generation). The Surinamese group was further classified according to self-reported ethnic origin into 'African', 'South-Asian' or other. Participants were considered of Dutch ethnic origin (henceforth, Dutch), if the participant and both parents were born in the Netherlands. Data for the HELIUS study were collected via the use of a questionnaire and a physical examination from January 2011 up to November 2015. For the current study, we used data of 23 942 participants for whom questionnaire data were available. Participants with an unknown/other ethnicity (n=50), Javanese Surinamese (n=250) or other/unknown Surinamese ethnicity (n=286) and those with missing data on smoking status (n=107), marital status (n=134) and/or educational level (n=186) were excluded, resulting in a total sample size of 22 929 participants, which was used in the current analyses.

### Smoking variables
Smoking status was determined by the question "Do you smoke at all?" Current smoking prevalence was calculated as the percentage of individuals in each ethnic group who responded 'yes' to this question. The prevalence of former smoking, also known as quit ratio per cent,[22] in each ethnic group was determined by dividing the number of ex-smokers (those who responded "no, but I used to smoke") by the number of ever-smokers (the sum of ex-smokers and current smokers).

### Socioeconomic status variables
The highest level of educational qualification attained (either in the Netherlands or in the country of origin) was grouped into three categories: 1) never been to school, or had elementary schooling only, or lower vocational schooling, or lower secondary schooling; 2) intermediate vocational schooling or higher secondary education schooling and 3) higher vocational schooling or university level education. Occupational level was classified according to Dutch Standard Occupational Classification system for 2010.[23] This document provides an extensive systematic list of all professions in the Dutch system. Based on this document, occupational level was classified into 'elementary', 'lower', 'intermediate', 'higher' or 'graduate', based on job title and job description, including a question on fulfilling an executive function. Occupation-related SES was created by combining employment status (unemployed, not in working force (retired, student or full-time homeworker), incapacitated or employed) with occupational

level (elementary, lower, middle, higher or graduate level) among those who were employed. This resulted in the following categories: 1) unemployed, 2) not in the workforce, 3) incapacitated, 4) employed at elementary/lower occupational level and 5) employed at intermediate to graduate occupational level. The sixth category represented participants who were employed, but there was insufficient data to define their occupational level. We grouped the elementary to lower level and intermediate to graduate level categories of the occupation-related SES variable, due to small numbers in these categories across ethnic groups.

## Statistical methods

Men and women were analysed separately since literature shows that smoking behaviour, including the usage and reasons for using tobacco and cessation, differs substantially by gender.[24] To make fair comparisons of the smoking prevalence across all ethnic groups, age-standardised prevalence rates for current smoking and former smoking were calculated using the entire HELIUS population as the standard population, stratified by 10-year age groups. A likelihood ratio test was used to test whether there were interactions between ethnicity and educational level and between ethnicity and occupation-related SES.

A Poisson regression model with a robust variance estimator was used to directly estimate adjusted prevalence ratios with 95% CIs for current smoking and former smoking in each ethnic minority group relative to the Dutch.[25] Prevalence ratios have been widely recommended in the literature as the preferred estimate if the outcome of interest is common as OR tends to greatly overestimate the risk ratio in cross-sectional studies.[25 26]

The SES of ethnic minority groups living in high-income countries may be lower than the native host population and the relationship between SES and risk factors for health is skewed in that lower SES groups have increased risk factor tendencies and worse health outcomes than the more economically advantaged groups.[27] Since we anticipated that SES may lie on the causal pathway between ethnicity and smoking behaviour,[20] we expect that the relationship between ethnicity and smoking will be mediated by SES. Therefore, we assessed the extent to which socioeconomic factors explained ethnic variations in current smoking and former smoking by adjusting ethnic differences in smoking and former smoking for educational level and occupation-related SES along with age and marital status. This was compared with the models adjusted for age and marital status only.

The presence of significant interactions between ethnicity and educational level (p<0.0001) and between ethnicity and occupation-related SES (p<0.0001) in both genders overall were observed. We therefore further stratified the analysis by ethnic group and gender to assess the relationship between SES and smoking prevalence and former smoking while adjusting for age and marital status. All analyses were performed using Stata V.14.1 and bar charts were plotted using Excel 2013.

## RESULTS

### Sociodemographic characteristics of study population

Table 1 shows the characteristics of the study population by gender and ethnic groups. On average, Moroccan and Turkish participants were younger than the Dutch, Ghanaian and Surinamese groups. Dutch men and women were more highly educated and had the lowest proportion of unemployed participants as compared with the ethnic minority groups.

Overall, the majority of ethnic minority groups were first-generation migrants (76.1%), and 23.9% were second-generation migrants. Ghanaians had the least amount of second-generation migrants (5.3%), followed by African Surinamese (17.4%) and South-Asian Surinamese (24.6%) while the Turkish (31.5%) and Moroccan (33.5%) groups comprised more second-generation migrants.

### Current smoking

Among men, smoking was more common in those of Turkish, African Surinamese and South-Asian Surinamese origin than Moroccan and Dutch origin, while Ghanaians smoked the least (figure 1). Among women, fewer Ghanaian and Moroccan women smoked compared with Dutch, Turkish, African Surinamese and South-Asian Surinamese women (figure 1). After adjustment for age and marital status, men of Turkish, African Surinamese and South-Asian Surinamese origin were significantly more likely than men of Dutch origin to be current smokers, whereas Moroccan men did not significantly differ from the Dutch (table 2, model 1). Additional adjustment for SES variables reduced the increased prevalence ratio of current smoking observed among Turkish, South-Asian Surinamese and African Surinamese men relative to Dutch men; however, they still smoked significantly more than the Dutch (table 2, model 4). The direction of the association changed for Moroccan men from 10% more likely to 18% less likely to smoke than Dutch men and the likelihood that Ghanaian men were less likely to smoke increased from 74% to 81% less likely (95% CI 85% to 75%) than Dutch men to be smokers.

Among women, only Turkish women were significantly more likely to be current smokers as compared with Dutch women, whereas Moroccan, South-Asian Surinamese and Ghanaian women were significantly less likely to be current smokers than women of Dutch origin (table 2, model 1). For women, after additional adjustment for SES, women from all ethnic minority groups except Turkish were significantly less likely to smoke compared with Dutch women. Turkish women did not significantly differ from Dutch women anymore, after adjustment for SES (table 2, models 4), whereas adjustment for SES made African Surinamese women significantly less likely to be smokers than Dutch women.

### Former smoking

Among both sexes, the highest prevalence of former smoking was in Ghanaians while the lowest was in

**Table 1** Socioeconomic characteristics of the study population by ethnicity and sex

| | Dutch | Moroccan | Turkish | African Surinamese | South-Asian Surinamese | Ghanaian |
|---|---|---|---|---|---|---|
| Total sample size n=22 929 (%) | n=4598 (20.1%) | n=4262 (18.6%) | n=3985 (17.4%) | n=4369 (19.1%) | n=3320 (14.5%) | n=2395 (10.5%) |
| **Men n=9745** | n=2115 | n=1601 | n=1800 | n=1764 | n=1536 | n=929 |
| Age, mean (SD) | 46.9 (13.8) | 41.5 (12.9) | 40.2 (12.4) | 47.5 (13.4) | 44.3 (13.8) | 46.2 (12.0) |
| **Education, n (%)** | | | | | | |
| Higher vocational/university | 1257 (59.4) | 297 (18.6) | 278 (15.4) | 312 (17.7) | 360 (23.4) | 85 (9.2) |
| Intermediate vocational/higher secondary | 497 (23.5) | 539 (33.7) | 527 (29.3) | 608 (34.5) | 484 (31.5) | 283 (30.5) |
| Lower vocational or secondary and below | 361 (17.1) | 765 (47.8) | 995 (55.3) | 844 (47.9) | 692 (45.1) | 561 (60.4) |
| **Occupation, n (%)** | | | | | | |
| Unemployed | 124 (5.9) | 271 (17.0) | 285 (16.1) | 323 (18.5) | 227 (15.0) | 158 (16.1) |
| Not in the working force* | 338 (16.0) | 127 (8.0) | 129 (7.3) | 195 (11.2) | 180 (11.9) | 66 (7.2) |
| Incapacitated† | 57 (2.7) | 166 (10.4) | 158 (8.9) | 132 (7.6) | 126 (8.3) | 51 (5.5) |
| Employed at elementary/lower occupational level | 186 (8.8) | 515 (32.3) | 681 (38.3) | 529 (30.4) | 389 (25.6) | 517 (56.1) |
| Employed at middle to graduate occupational level | 1370 (64.8) | 452 (28.4) | 462 (26.0) | 510 (29.3) | 555 (36.6) | 94 (10.2) |
| Employed but unknown occupational level‡ | 38 (1.8) | 63 (4.0) | 61 (3.4) | 54 (3.1) | 41 (2.7) | 35 (3.8) |
| **Women n=13 184** | n=2483 | n=2661 | n=2185 | n=2605 | n=1784 | n=1466 |
| Age, mean (SD) | 45.4 (14.2) | 38.6 (13.0) | 39.5 (12.4) | 47.5 (12.4) | 45.7 (13.3) | 42.9 (11.0) |
| **Education, n (%)** | | | | | | |
| Higher vocational/university | 1522 (61.3) | 439 (16.5) | 304 (13.9) | 672 (25.8) | 390 (21.9) | 66 (4.5) |
| Intermediate vocational/higher secondary | 516 (20.8) | 633 (29.0) | 923 (34.7) | 958 (36.8) | 513 (28.8) | 333 (22.7) |
| Lower vocational or secondary and below | 445 (17.9) | 1299 (48.8) | 1248 (57.1) | 975 (37.4) | 881 (49.4) | 1067 (72.8) |
| **Occupation, n (%)** | | | | | | |
| Unemployed | 130 (5.3) | 389 (14.8) | 300 (13.9) | 385 (14.9) | 277 (15.7) | 408 (28.4) |
| Not in the working force* | 465 (18.8) | 1048 (39.9) | 792 (36.8) | 323 (12.5) | 297 (16.8) | 115 (8.0) |
| Incapacitated† | 84 (3.4) | 177 (6.7) | 199 (9.2) | 267 (10.3) | 179 (10.1) | 161 (11.2) |
| Employed at elementary/lower occupational level | 240 (9.7) | 349 (13.3) | 380 (17.6) | 417 (16.1) | 371 (21.0) | 611 (42.6) |
| Employed at middle to graduate occupational level | 1532 (61.9) | 607 (23.1) | 615 (34.8) | 1121 (43.4) | 615 (34.8) | 104 (7.3) |
| Employed but unknown occupational level‡ | 25 (1.0) | 59 (2.2) | 31 (1.8) | 73 (2.8) | 31 (1.8) | 36 (2.5) |

*Retired or student or full-time homeworker.
†disabled or unfit for work.
‡Working but did not provide sufficient information when describing their jobs.

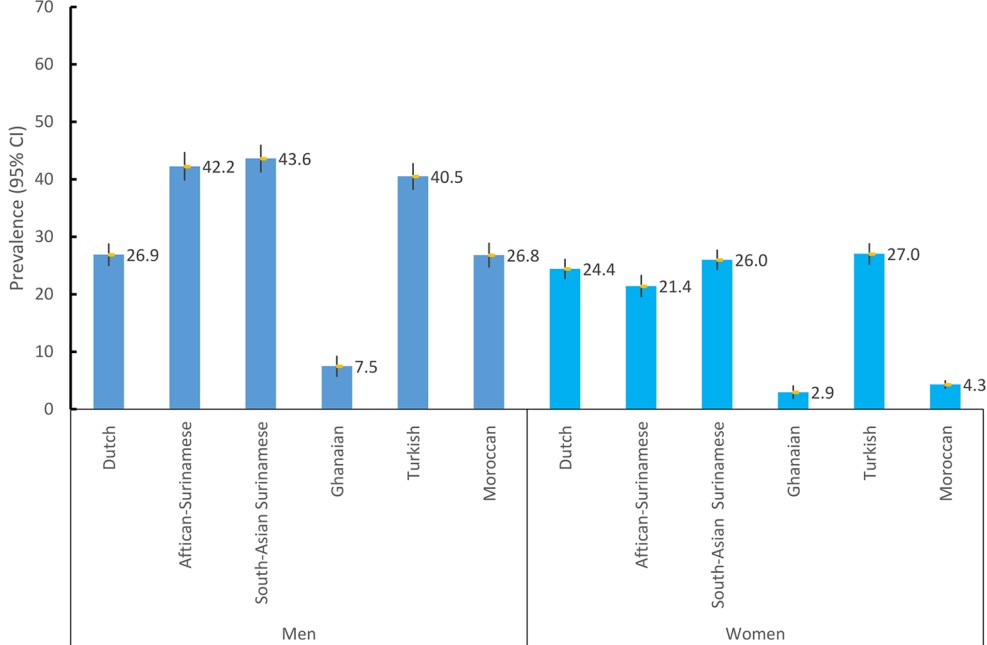

**Figure 1** Age-standardised prevalence of current smoking by gender and ethnicity.

African Surinamese men and Turkish women (figure 2). All ethnic minority groups were significantly less likely to quit smoking as compared with Dutch, except for Ghanaians who were more likely to have quit (although not statistically significant in men) (table 3, model 1). For former smoking, although the Turkish, Surinamese and Moroccan ethnic groups appeared to be less likely to quit smoking than the Dutch, the lower likelihoods were partly explained by socioeconomic factors (table 3, model 4). Ghanaians were even more likely to have a higher former smoking rate than the Dutch after socioeconomic factors were accounted for.

### The relationship between SES and smoking prevalence and former smoking within each ethnic group

Within each ethnic group, we observed SES inequalities in smoking prevalence (table 4). For both sexes, participants with higher educational levels were less likely to smoke than those with lower educational levels in all ethnic groups, although not always statistically significant. Although very wide confidence intervals were obtained for Ghanaian women, those with higher vocational schooling or university level education smoked more than women with little or no education.

Compared with those employed in middle to graduate level positions, all unemployed men smoked more in all ethnic groups. Although not always statistically significant, all unemployed and lower level professional women except Ghanaians were more likely to smoke. Persons who were disabled or unfit for work (incapacitated) also had a higher prevalence of smoking in all ethnic groups, although not statistically significant.

Within each ethnic group, SES inequalities were observed in former smoking (table 5). For both sexes, participants with higher levels of education were more

likely to quit smoking, although not significant for Moroccan men and women. Higher educated Ghanaian women seemed less likely to quit smoking, yet this was not significant. For both sexes, unemployed participants were less likely to quit compared with the middle or graduate level participants. Participants employed in lower level professional jobs were less likely to quit than those employed at the middle or graduate level except for Ghanaian women who appeared to be more likely to quit. A similar lower likelihood of quitting was observed among the incapacitated with the exception of Moroccan women.

## DISCUSSION
### Key findings

The prevalence of current smoking and former smoking of adult men and women of minority ethnic groups living in Amsterdam is different from the Dutch host population. Accounting for socioeconomic factors, the analysis showed that Turkish, South-Asian Surinamese and African Surinamese men smoked more than Dutch men, while Ghanaian and Moroccan men smoked less. The smoking prevalence of Turkish women was similar to Dutch women, while Ghanaian, Moroccan, South-Asian Surinamese and African Surinamese women smoked less. Only the Ghanaian ethnic group was more likely to quit than the Dutch. Socioeconomic factors influenced current smoking and former smoking prevalence, but more so among men than among women and differently depending on the ethnic minority group. SES contributed to the higher prevalence of smoking among Turkish, Surinamese and Moroccan men and Turkish women, but diminished the estimates for Surinamese, Moroccan and Ghanaian women who were less likely to smoke than the

**Table 2** Prevalence and adjusted prevalence ratios (PRs) with 95% CIs of current smoking of ethnic minority groups compared with Dutch

| Ethnic group | Crude prevalence n/N (%) | Model 0 PR (95% CI) | Model 1* PR (95% CI) | Model 2† PR (95% CI) | Model 3‡ PR (95% CI) | Model 4§ PR (95% CI) |
|---|---|---|---|---|---|---|
| **Men** | | | | | | |
| Dutch | 562/2115 (26.6) | 1.00 | 1.00 | 1.00 | 1.00 | 1.00 |
| Turkish | 772/1800 (42.9) | 1.57 (1.44 to 1.72) | 1.69 (1.54 to 1.86) | 1.33 (1.20 to 1.46) | 1.33 (1.20 to 1.47) | 1.23 (1.11 to 1.36) |
| African Surinamese | 780/1764 (44.2) | 1.68 (1.54 to 1.84) | 1.55 (1.41 to 1.69) | 1.26 (1.15 to 1.39) | 1.28 (1.17 to 1.41) | 1.20 (1.09 to 1.32) |
| South-Asian Surinamese | 645/1536 (42.0) | 1.57 (1.43 to 1.72) | 1.53 (1.40 to 1.68) | 1.27 (1.15 to 1.39) | 1.29 (1.17 to 1.42) | 1.21 (1.10 to 1.33) |
| Moroccan | 446/1601 (27.9) | 1.02 (0.92 to 1.14) | 1.10 (0.99 to 1.23) | 0.88 (0.79 to 0.99) | 0.87 (0.78 to 0.98) | 0.82 (0.73 to 0.92) |
| Ghanaian | 69/929 (7.4) | 0.28 (0.22 to 0.36) | 0.26 (0.20 to 0.33) | 0.20 (0.16 to 0.25) | 0.20 (0.16 to 0.26) | 0.19 (0.15 to 0.24) |
| **Women** | | | | | | |
| Dutch | 593/2483 (23.9) | 1.00 | 1.00 | 1.00 | 1.00 | 1.00 |
| Turkish | 647/2185 (29.6) | 1.15 (1.05 to 1.27) | 1.26 (1.14 to 1.40) | 1.07 (0.96 to 1.20) | 1.09 (0.98 to 1.21) | 1.03 (0.92 to 1.15) |
| African Surinamese | 660/2605 (25.3) | 1.09 (0.99 to 1.21) | 0.95 (0.86 to 1.05) | 0.79 (0.71 to 0.88) | 0.81 (0.74 to 0.90) | 0.75 (0.68 to 0.83) |
| South-Asian Surinamese | 370/1784 (20.7) | 0.88 (0.78 to 0.98) | 0.82 (0.73 to 0.92) | 0.68 (0.61 to 0.77) | 0.70 (0.62 to 0.79) | 0.65 (0.58 to 0.73) |
| Moroccan | 145/2661 (5.5) | 0.21 (0.18 to 0.25) | 0.23 (0.19 to 0.27) | 0.19 (0.16 to 0.23) | 0.20 (0.16 to 0.24) | 0.19 (0.16 to 0.23) |
| Ghanaian | 37/1466 (2.5) | 0.10 (0.07 to 0.14) | 0.09 (0.06 to 0.12) | 0.07 (0.05 to 1.20) | 0.07 (0.05 to 0.10) | 0.06 (0.05 to 0.09) |

Model 0: adjusted for age only.
*Model 1: adjusted for age and marital status.
†Model 2: adjusted for age, marital status and education.
‡Model 3: adjusted for age, marital status and occupation-related SES.
§Model 4: adjusted for age, marital status, education and occupational-related SES.
SES, socioeconomic status.

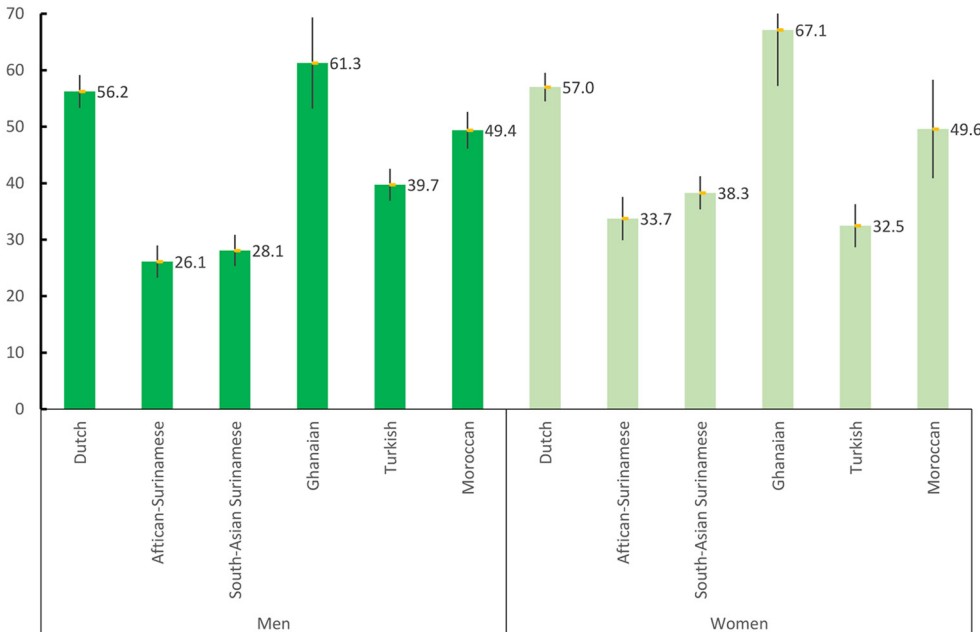

**Figure 2** Age-standardised prevalence of former smoking by gender and ethnicity.

Dutch. Socioeconomic factors also accounted for part of the lower prevalence of former smoking observed among the ethnic minority groups relative to the Dutch.

### Discussion of key findings

Smoking behaviour differed among the various ethnic minority groups compared with the Dutch. The prevalence of smoking among Ghanaians in the Netherlands has not been previously studied. Comparable studies in Amsterdam, which used the same country of birth criteria for ethnicity, indicates that smoking prevalence among adults aged 35–60 years has reduced over the last decade (from 2000/2003 to 2011/2015) in Moroccan, Turkish and Surinamese groups.[11] Age-standardised estimates of smoking prevalence among Turkish men (63%) declined to ~41.7%, and from 55% to 44.8% in Surinamese men, but remained fairly constant in Moroccan men (~29%). However, a slight increase in smoking prevalence from <1% to ~2.6% among Moroccan women was observed. The presence of stronger tobacco control policies in the Netherlands after 2001 may be a contributing factor for the decline in smoking prevalence among Turkish and Surinamese men over the last decade bringing it closer to Dutch men.[28]

When referring to estimates of smoking from population-based studies conducted in the respective countries of origin, we observed more variability between women in the home country versus women in Amsterdam: 13% in Turkey vs 27% (Turkish),[29] 9.9% in Suriname vs 24.2% (Surinamese),[30] 3.3% in Morocco vs 4.3% (Moroccan)[31] and 0.3% in Ghana vs 2.9% (Ghanaian).[32] Whereas for men, closer parallels between men in their home countries and men in Amsterdam: 41% vs 40.5% (Turkish),[29] 38.4% vs 43.2% (Surinamese),[30] 31.5% vs 26.8% (Moroccan)[31] and 8.9% vs 7.5% (Ghanaian)

were detected.[32] These differences, especially among women, allude to the possible impact of acculturation, and different social norms on smoking.[8–10 33] Focusing targeted ethnic smoking interventions on people at the same time they immigrate maybe a potential strategy that can be used for smoking prevention and reduction.

From our findings, lower SES was related to higher smoking prevalence and lower former smoking more in men in most ethnic groups and is consistent with findings from past research.[34–36] Smoking was more common among lower educated men of Turkish and host populations in a study conducted in Germany and the Netherlands.[37] However, higher educated Turkish women smoked more than lower educated, while the reverse was evident in the host population.[37] Higher smoking among the more educated women was observed only in Ghanaian women but very wide CIs were obtained due to small numbers. Higher smoking among the more educated women in Amsterdam may flag the signs of the early stages of tobacco smoking epidemic among Ghanaian women,[38 39] who may be challenging cultural norms and taboos towards smoking present especially among women in the Ghanaian community.

For former smoking, higher prevalences among the Dutch than African and South-Asian Surinamese are known.[17] This study confirmed that Turkish and Surinamese smokers have lower former smoking prevalence, while Ghanaian smokers are more likely to quit than the Dutch. There were indications that Moroccan and Dutch had similar quit ratios. Others factors including lack of social support, more stressful lives or living in deprived neighbourhoods may be ways in which lower SES indirectly influences smoking cessation.[40]

**Table 3** Prevalence and adjusted prevalence ratios (PRs) with 95% CIs of former smoking of ethnic minority groups compared with Dutch

### Former smoking/quit ratio per cent

| Ethnic group | Crude prevalence n/N (%) | Model 0 PR (95% CI) | Model 1* PR (95% CI) | Model 2† PR (95% CI) | Model 3‡ PR (95% CI) | Model 4§ PR (95% CI) |
|---|---|---|---|---|---|---|
| **Men** | | | | | | |
| Dutch | 827/1389 (59.5) | 1.00 | 1.00 | 1.00 | 1.00 | 1.00 |
| Turkish | 442/1214 (36.4) | 0.70 (0.64 to 0.76) | 0.67 (0.62 to 0.74) | 0.76 (0.71 to 0.85) | 0.78 (0.71 to 0.85) | 0.82 (0.74 to 0.90) |
| African Surinamese | 363/1143 (31.8) | 0.52 (0.47 to 0.57) | 0.57 (0.52 to 0.63) | 0.63 (0.57 to 0.70) | 0.63 (0.57 to 0.70) | 0.65 (0.59 to 0.72) |
| South-Asian Surinamese | 239/884 (27.0) | 0.48 (0.43 to 0.54) | 0.50 (0.45 to 0.56) | 0.56 (0.50 to 0.63) | 0.56 (0.50 to 0.63) | 0.58 (0.51 to 0.65) |
| Moroccan | 432/878 (49.2) | 0.90 (0.84 to 0.98) | 0.88 (0.82 to 0.95) | 1.00 (0.92 to 1.09) | 1.00 (0.92 to 1.09) | 1.06 (0.97 to 1.16) |
| Ghanaian | 118/187 (63.1) | 1.02 (0.91 to 1.15) | 1.08 (0.96 to 1.22) | 1.23 (1.09 to 1.40) | 1.23 (1.09 to 1.40) | 1.34 (0.74 to 0.90) |
| **Women** | | | | | | |
| Dutch | 918/1511 (60.8) | 1.00 | 1.00 | 1.00 | 1.00 | 1.00 |
| Turkish | 253/900 (28.1) | 0.54 (0.48 to 0.61) | 0.54 (0.48 to 0.61) | 0.66 (0.58 to 0.74) | 0.60 (0.53 to 0.67) | 0.66 (0.59 to 0.75) |
| African Surinamese | 470/1130 (41.6) | 0.68 (0.63 to 0.74) | 0.75 (0.69 to 0.81) | 0.84 (0.77 to 0.92) | 0.79 (0.72 to 0.85) | 0.85 (0.78 to 0.92) |
| South-Asian Surinamese | 193/563 (34.3) | 0.60 (0.53 to 0.68) | 0.64 (0.56 to 0.72) | 0.73 (0.64 to 0.82) | 0.68 (0.60 to 0.76) | 0.73 (0.65 to 0.83) |
| Moroccan | 93/238 (39.1) | 0.83 (0.70 to 0.97) | 0.87 (0.74 to 1.03) | 1.00 (0.85 to 1.18) | 0.94 (0.80 to 1.11) | 1.01 (0.86 to 1.19) |
| Ghanaian | 75/112 (67.0) | 1.12 (0.97 to 1.29) | 1.24 (1.07 to 1.43) | 1.53 (1.31 to 1.77) | 1.43 (1.22 to 1.67) | 1.59 (1.36 to 1.86) |

Model 0: adjusted for age only.
*Model 1: adjusted for age and marital status.
†Model 2: adjusted for age, marital status and education.
‡Model 3: adjusted for age, marital status and occupation-related SES.
§Model 4: adjusted for age, marital status, education and occupational-related SES.
SES, socioeconomic status.

**Table 4** Current smoking PRs (95% CI) among men and women of different educational levels and occupation-related SES within each ethnic group

| Indicator of SES | Dutch | South-Asian Surinamese | African Surinamese | Turkish | Moroccan | Ghanaian |
|---|---|---|---|---|---|---|
| | PR (95% CI)* | PR (95% CI)* | PR (95% CI)* | PR (95% CI)* | PR (95% CI)* | PR (95% CI)* |
| **Men** | | | | | | |
| *Education* | | | | | | |
| Higher vocational school/university | 0.57 (0.47 to 0.69) | 0.44 (0.36 to 0.53) | 0.65 (0.55 to 0.78) | 0.50 (0.41 to 0.62) | 0.62 (0.49 to 0.80) | 0.28 (0.07 to 1.19) |
| Intermediate vocational/higher secondary | 0.57 (0.57 to 0.87) | 0.69 (0.61 to 0.80) | 0.92 (0.82 to 1.03) | 0.83 (0.73 to 0.94) | 0.83 (0.70 to 0.99) | 0.78 (0.46 to 1.33) |
| Lower vocational and below | 1.00 | 1.00 | 1.00 | 1.00 | 1.00 | 1.00 |
| *Occupation-related SES* | | | | | | |
| Unemployed | 1.91 (1.53 to 2.38) | 1.69 (1.43 to 1.99) | 1.58 (1.35 to 1.85) | 1.32 (1.11 to 1.56) | 1.81 (1.46 to 2.26) | 5.10 (1.17 to 22.30) |
| Not in the working force† | 1.11 (0.87 to 1.42) | 0.99 (0.76 to 1.29) | 1.22 (0.96 to 1.54) | 0.73 (0.53 to 1.02) | 0.60 (0.36 to 1.01) | 2.37 (0.37 to 15.33) |
| Incapacitated‡ | 1.58 (1.10 to 2.30) | 1.30 (1.02 to 1.65) | 1.48 (1.21 to 1.81) | 1.11 (0.88 to 1.41) | 1.49 (1.11 to 2.00) | 4.34 (0.83 to 22.77) |
| Employed at elementary/lower occupational level | 1.50 (1.20 to 1.87) | 1.57 (1.34 to 1.84) | 1.32 (1.13 to 1.53) | 1.27 (1.10 to 1.46) | 1.33 (1.07 to 1.64) | 3.37 (0.79 to 14.37) |
| Employed at middle to graduate occupational level | 1.00 | 1.00 | 1.00 | 1.00 | 1.00 | 1.00 |
| **Women** | | | | | | |
| *Education* | | | | | | |
| Higher vocational school/university | 0.39 (0.33 to 0.46) | 0.61 (0.47 to 0.80) | 0.58 (0.48 to 0.70) | 0.72 (0.56 to 0.91) | 0.87 (0.53 to 1.45) | 6.02 (2.23 to 16.25) |
| Intermediate vocational/higher secondary | 0.60 (0.50 to 0.73) | 0.87 (0.70 to 1.08) | 0.75 (0.65 to 0.87) | 1.01 (0.87 to 1.17) | 0.90 (0.59 to 1.37) | 0.56 (0.19 to 1.62) |
| Lower vocational and below | 1.00 | 1.00 | 1.00 | 1.00 | 1.00 | 1.00 |
| *Occupation-related SES* | | | | | | |
| Unemployed | 1.27 (0.96 to 1.67) | 1.66 (1.30, 2.12) | 1.54 (1.29 to 1.84) | 1.16 (0.94 to 1.43) | 1.42 (0.90 to 2.22) | 0.70 (0.18 to 2.68) |
| Not in the working force† | 1.09 (0.86 to 1.38) | 1.04 (0.74 to 1.45) | 1.11 (0.86 to 1.45) | 1.03 (0.85 to 1.25) | 0.86 (0.53 to 1.39) | 1.44 (0.33 to 6.25) |
| Incapacitated‡ | 1.38 (1.01 to 1.90) | 1.82 (1.34 to 2.47) | 1.47 (1.17 to 1.84) | 1.22 (0.95 to 1.56) | 1.40 (0.77 to 2.57) | 0.76 (0.17 to 3.49) |
| Employed at elementary/lower occupational level | 1.69 (1.39 to 2.07) | 1.23 (0.94 to 1.61) | 1.34 (1.11 to 1.62) | 1.29 (1.06 to 1.57) | 1.22 (0.75 to 1.99) | 0.44 (0.12 to 1.67) |
| Employed at middle to graduate occupational level | 1.00 | 1.00 | 1.00 | 1.00 | 1.00 | 1.00 |

*PR adjusted for age and marital status.
†Retired or student or full-time homeworker.
‡Disabled or unfit for work.
PR, prevalence ratios; SES, socioeconomic status.

**Table 5** Former smoking prevalence ratios (PRs) (95% CI) among men and women of different educational levels and occupation-related SES within each ethnic group

| Indicator of SES | Dutch | South-Asian Surinamese | African Surinamese | Turkish | Moroccan | Ghanaian |
|---|---|---|---|---|---|---|
| | PR (95% CI)* | PR (95% CI)* | PR (95% CI)* | PR (95% CI)* | PR (95% CI)* | PR (95% CI)* |
| **Men** | | | | | | |
| *Education* | | | | | | |
| Higher vocational school/university | 1.28 (1.14 to 1.45) | 1.84 (1.44 to 2.37) | 1.38 (1.12 to 1.69) | 1.54 (1.27 to 1.87) | 1.16 (0.95 to 1.41) | 1.38 (1.05 to 1.82) |
| Intermediate vocational/higher secondary | 1.19 (1.04 to 1.37) | 1.09 (0.82 to 1.44) | 0.97 (0.80 to 1.19) | 1.12 (0.94 to 1.34) | 1.02 (0.88 to 1.19) | 1.09 (0.84 to 1.41) |
| Lower vocational and below | 1.00 | 1.00 | 1.00 | 1.00 | 1.00 | 1.00 |
| *Occupation-related SES* | | | | | | |
| Unemployed | 0.71 (0.56 to 0.89) | 0.59 (0.41 to 0.86) | 0.58 (0.44 to 0.76) | 0.85 (0.67 to 1.08) | 0.77 (0.61 to 0.96) | 0.66 (0.47 to 0.91) |
| Not in the working force† | 1.02 (0.91 to 1.14) | 0.91 (0.64 to 1.31) | 0.81 (0.62 to 1.07) | 0.89 (0.60 to 1.32) | 0.99 (0.75 to 1.29) | 0.65 (0.36 to 1.15) |
| Incapacitated‡ | 0.83 (0.63 to 1.09) | 0.79 (0.52 to 1.17) | 0.68 (0.48 to 0.97) | 0.96 (0.74 to 1.23) | 0.88 (0.70 to 1.09) | 0.77 (0.51 to 1.18) |
| Employed at elementary/lower occupational level | 0.82 (0.69 to 0.98) | 0.57 (0.43 to 0.76) | 0.72 (0.58 to 0.89) | 0.85 (0.70 to 1.03) | 0.93 (0.78 to 1.11) | 0.63 (0.47 to 0.84) |
| Employed at middle to graduate occupational level | 1.00 | 1.00 | 1.00 | 1.00 | 1.00 | 1.00 |
| **Women** | | | | | | |
| *Education* | | | | | | |
| Higher vocational school/university | 1.71 (1.52 to 1.93) | 1.99 (1.33 to 2.31) | 1.31 (1.10 to 1.55) | 1.73 (1.33 to 2.25) | 1.09 (0.71 to 1.66) | 0.72 (0.36 to 1.41) |
| Intermediate vocational/higher secondary | 1.41 (1.21 to 1.62) | 1.75 (1.33 to 2.31) | 1.18 (1.00 to 1.38) | 1.21 (0.95 to 1.56) | 1.12 (0.78 to 1.61) | 1.28 (0.98 to 1.65) |
| Lower vocational and below | 1.00 | 1.00 | 1.00 | 1.00 | 1.00 | 1.00 |
| *Occupation-related SES* | | | | | | |
| Unemployed | 0.89 (0.73 to 1.08) | 0.62 (0.44 to 0.88) | 0.73 (0.58 to 0.92) | 0.66 (0.48 to 0.91) | 0.97 (0.61 to 1.53) | 0.95 (0.60 to 1.51) |
| Not in the working force† | 0.92 (0.83 to 1.03) | 0.77 (0.55 to 1.07) | 0.98 (0.79 to 1.20) | 0.54 (0.40 to 0.73) | 1.02 (0.64 to 1.62) | 0.52 (0.22 to 1.22) |
| Incapacitated‡ | 0.86 (0.69 to 1.08) | 0.57 (0.38 to 0.87) | 0.85 (0.69 to 1.05) | 0.65 (0.45 to 0.96) | 1.20 (0.75 to 1.91) | 0.97 (0.55 to 1.71) |
| Employed at elementary/lower occupational level | 0.76 (0.65 to 0.89) | 0.48 (0.32 to 0.73) | 0.76 (0.61 to 0.94) | 0.60 (0.43 to 0.82) | 0.84 (0.49 to 1.44) | 1.07 (0.69 to 1.66) |
| Employed at middle to graduate occupational level | 1.00 | 1.00 | 1.00 | 1.00 | 1.00 | 1.00 |

*PR adjusted for age and marital status.
†Retired or student or full-time homeworker.
‡Disabled or unfit for work.
PR, prevalence ratios; SES, socioeconomic status.

Higher rates of smoking were observed among both men and women employed in lower level professional jobs than those in higher professional jobs of all ethnic groups except Ghanaian and Moroccan men in which the difference may not have been detected due to small numbers. Enforcing restrictions on smoking on the workplace can make it more difficult for even workers in lower level jobs to smoke on the job.[41]

Although SES contributed to observed ethnic differences in smoking and cessation, it did not entirely explain the ethnic differences suggesting that other factors may play a role. A possible explanation is that smoking behaviour may be influenced by cultural norms present in the ethnic group, as those sharing a particular ethnic identity tend to share similar attitudes towards smoking.[42]

The strengths of the HELIUS study is that it used large sample sizes in each ethnic group and participants were recruited in a systematic manner to reflect a representative sample of adults in each ethnic group living in Amsterdam.[20] Smoking status may be under-reported since biochemical methods were not used to verify self-reported smoking statuses. People tend to under-report undesirable health behaviours, especially those stigmatised within the wider society.[43] Urine analysis of cotinine levels, the main metabolite of nicotine, is a useful biochemical indicator that can be used to validate current smoking among participants.[44] Our research did not focus on analysing what current smokers smoked; however, the majority used cigarettes rather than cigars and packages of pipe tobacco. Smoking cessation was assessed by using quit ratios, which is a cumulative measure of a number of former smokers in the ethnic group, which occurred over time and not necessarily an indicator of a recent smoking cessation or assessment at the individual level.[41]

## CONCLUSION

Current smoking and former smoking varies by ethnic group in the Netherlands. The higher levels of smoking prevalence and the lower levels of former smoking among some ethnic minority groups compared with the Dutch are largely, but not completely, explained by socioeconomic factors. Although differences were less marked after adjustment, particularly for socioeconomic factors, the findings still suggest that some ethnic minority groups may benefit from targeted cessation interventions.

**Acknowledgements** The Healthy Life in an Urban Setting (HELIUS) study is conducted by the Academic Medical Center Amsterdam and the Public Health Service of Amsterdam. Both organisations provided core support for HELIUS. We are most grateful to the participants of the HELIUS study and the management team, research nurses, interviewers, research assistants and other staff who have taken part in gathering the data of this study.

**Contributors** The study idea was conceived by RB in collaboration with LS, JA and CA. The first draft of the paper was written by RB. LS, JA, CA, EMD, AEK, RP and MBS, all wrote and contributed to several updated versions.

**Funding** The HELIUS study is funded by the Dutch Heart Foundation (Hartstichting) (grant number 2010T084), the Netherlands Organization for Health Research and Development (ZonMw grant number 200500003), and the European Commission Seventh Framework Programme (FP-7) (grant number 278901). The views

expressed in the paper are those of the authors, the funders had no participation in the writing of this paper.

**Competing interests** None declared.

**Patient consent** Detail has been removed from this case description/these case descriptions to ensure anonymity. The editors and reviewers have seen the detailed information available and are satisfied that the information backs up the case the authors are making.

**Ethics approval** Institutional Review Board of the Academic Medical Center at the University of Amsterdam.

**Provenance and peer review** Not commissioned; externally peer reviewed.

**Data sharing statement** There is no additional unpublished data from the study.

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
