## [Reviewer comments · BMJ Open]

ARTICLE DETAILS

TITLE (PROVISIONAL)	Ethnic differences in current smoking and former smoking in the Netherlands and the contribution of socio-economic factors: a cross-sectional analysis of the HELIUS Study
AUTHORS	Brathwaite, Rachel; Smeeth, Liam; Addo, Juliet; Kunst, Anton; Peters, Ron; Snijder, Marieke; Derks, Eske; Agyemang, Charles

VERSION 1 - REVIEW

REVIEWER	Chung-Il Wi, MD Assistant Professor of Pediatrics Mayo Clinic USA
REVIEW RETURNED	24-Feb-2017

GENERAL COMMENTS	This study aimed 1) to investigate difference in prevalence of current and former smoking rate by different gender, ethnicity and SES, and 2) to explain the extent to which this difference is explained by SES. Authors well described what is lacking in prevalence in smoking in Netherlands and tried to contribute to developing target strategy for reducing smoking rate by disentangling smoking pattern related to gender, ethnicity, and SES in Netherlands using prospective data. This study is timely and addresses an important need given the great appreciation for understanding ethnic minorities' health issue, especially how SES interact with ethnicity with regard to smoking. The reviewer has a few comments which are addressable before publication. 1. For the Title, the reviewer suggests adding "current". (ie, Ethnic differences in current smoking....)2. In the Method, authors described clearly how they defined key variables (ie, ethnicity, SES, and smoking variables) and how they analyzed given the interaction between ethnicity and SES. However, when categorizing occupation level by combining elementary and lower vs. intermediate to graduate, the reviewer wonders what is rationale for this subgrouping. It would be great if authors cite some references (eg, literature using same sub-grouping) or provide rationale based on sensitivity analysis.3. Although authors adjusted for marital status in the Models,
--

	rationale (citation or univariate analysis) was not provided. Are there any other variables available to be considered as confounders such as comorbidities (eg, asthma, depression)? 4. It would be informative if overall % of each ethnicity in Amsterdam is reported. 5. In the Model 2 of the Table 2, authors adjusted for both education and occupation-related SES which may cause over-adjustment given the collinearity between education and occupation. The reviewer suggests adjusting for each SES variable and report separately and jointly (eg, Model 2, 3, and 4) 6. Although 1st and 2nd generation among minorities were treated under same category, the reviewer is wondering if there is any difference between two generations. This data is not necessarily to be reported in the table, but the reviewer would recommend briefly describing this data in the text if available. 7. Lack of data on amount (or duration) of cigarette smoking in addition to type of smoking is also limitation.
--	--

REVIEWER	Stephen Thielke University of Washington, Seattle, WA, USE
REVIEW RETURNED	13-Mar-2017

GENERAL COMMENTS	This is interesting research, and it is important to characterize the epidemiology of smoking and smoking cessation. The methods were adequate to answer the scientific questions. My primary concern is about the interpretation of the results. This is a descriptive paper, with no intervention, and so one needs to be particularly careful about how to use the findings to inform policy. This appears in a number of ways: 1. You found an association between ethnic group status and sociodemographic status. You described this as follows: "SES contributed to the higher prevalence of smoking among Turkish, Surinamese and Moroccan men and Turkish women, but diminished the estimates for Surinamese, Moroccan and Ghanaian women who were less likely to smoke than the Dutch." What do you mean, "SES contributed"? This implies that somehow SES is the driver for smoking behavior independent of ethnic status, and suggests that SES is the cause of smoking. This just does not make sense. Instead, it appears that various ethnic groups have different rates of smoking and cessation, and these groups also have low SES. But there is no reason to believe that the SES itself was a cause of greater smoking. If you are in fact drawing this conclusion, you need to justify it in more detail. My own interpretation would be a bit more concrete: certain groups
--

	defined by ethnic status and sex have higher and lower rates of smoking, and might benefit from targeted stop-smoking interventions. 2. While it is generally known that smoking is unhealthy, there is not necessarily a guaranteed and uniform benefit from cessation at different age groups. Your discussion about "increasing hazard ratios for acute myocardial infarction and stroke across medium and lower SES ethnic groups compared to the higher SES groups" is thus very speculative. The claim that "reducing smoking and or increasing cessation among ethnic minority groups can help reduce the high all-cause mortality and cardiovascular disease incidence already seen among Turkish and Surinamese in the Netherlands" sounds plausible, but is also outside the scope of your research (since you did not examine any effects of smoking). Invoking both SES and ethnic status adds to the confusion. I recommend removing this entire section. 3. The home country vs immigrant rates were an interesting result, which deserved more attention and consideration. At the bottom of page 16, it was unclear which were the home country rates and which were the Amsterdam rates. I think the home country was the first one. The percentages did not correspond to the prevalence values in Table 2, so I was a bit confused. (And why is this comparison with people in Amsterdam instead of the entire sample?) This finding deserves more attention, since it points to a potential prevention approach, such as "catching" people at the time that they immigrate. (This would be true if more ethnic minority individuals smoked after immigrating, which I think is the case.) Why do you think, based on your findings, that immigrants smoke at different rates? (Potentially it is mediated by SES, but empirically answering that question would depend on knowing SES before and after immigration.)
--	--

VERSION 1 – AUTHOR RESPONSE

Reviewer #1:

This study aimed 1) to investigate difference in prevalence of current and former smoking rate by different gender, ethnicity and SES, and 2) to explain the extent to which this difference is explained by SES. Authors well described what is lacking in prevalence in smoking in Netherlands and tried to contribute to developing target strategy for reducing smoking rate by disentangling smoking pattern related to gender, ethnicity, and SES in Netherlands using prospective data. This study is timely and addresses an important need given the great appreciation for understanding ethnic minorities' health issue, especially how SES interact with ethnicity with regard to smoking. The reviewer has a few comments which are addressable before publication.

1. For the Title, the reviewer suggests adding "current". (ie, Ethnic differences in current smoking....) "Current" has now been included in the title as suggested.

2. In the Method, authors described clearly how they defined key variables (ie, ethnicity, SES, and smoking variables) and how they analyzed given the interaction between ethnicity and SES. However, when categorizing occupation level by combining elementary and lower vs. intermediate to graduate, the reviewer wonders what is rationale for this subgrouping. It would be great if authors cite some references (eg, literature using same sub-grouping) or provide rationale based on sensitivity analysis.

The rationale for grouping occupation level by elementary and lower vs intermediate to graduate was due to small numbers in the lower, middle, high and professional/graduate level categories within certain ethnic groups. This rationale has now been included in the methods section of the manuscript on page 7.

“We grouped the elementary to lower and intermediate to graduate level categories of the occupation-related SES variable, due to small numbers in these categories across ethnic groups.”

3. Although authors adjusted for marital status in the Models, rationale (citation or univariate analysis) was not provided. Are there any other variables available to be considered as confounders such as comorbidities (eg, asthma, depression)?

We have included a step by step adjustment for age, then age and marital status in new Tables 2 and 3. When marital status was added to the model (Models 1) it resulted in a change in the prevalence ratios for both smoking and former smoking for most ethnic groups. Marital status was associated with smoking for both men and women, as well as with ethnicity. We did not adjust for comorbidities such as asthma since it may be the result of smoking and therefore inappropriate to be considered as a confounder.

4. It would be informative if overall % of each ethnicity in Amsterdam is reported.

The overall percentage of each ethnic group in the HELIUS study is now included in Table 1

5. In the Model 2 of the Table 2, authors adjusted for both education and occupation-related SES which may cause over-adjustment given the collinearity between education and occupation. The reviewer suggests adjusting for each SES variable and report separately and jointly (eg, Model 2, 3, and 4)

As suggested, we have now adjusted separately for each variable: Age only in Model 0; Age and marital status in Model 1; age, marital status and education only in Model 2; age, marital status and occupation-related SES in Model 3 and all in Model 4. In addition, we have split this table into 2 tables (new Table 2 for current smoking and Table 3 for former smoking results) since it was now too large to accommodate the new results.

6. Although 1st and 2nd generation among minorities were treated under same category, the reviewer is wondering if there is any difference between two generations. This data is not necessarily to be reported in the table, but the reviewer would recommend briefly describing this data in the text if available.

A brief description of first and second generation migrants is now included in a paragraph on page 9 under the section for socio-demographic characteristics of the study population.

“Overall, the majority of ethnic minority groups were first generation migrants (76.1%), and 23.9% second generation migrants. However, the Ghanaian ethnic group had the least amount of second generation migrants (5.3%), followed by African Surinamese (17.4%) and South-Asian Surinamese (24.6%) while the Turkish (31.5%) and Moroccan (33.5%) groups comprised of more second generation migrants.”

7. Lack of data on amount (or duration) of cigarette smoking in addition to type of smoking is also a limitation.

We did not include this data, since it may be subject to lack of accuracy due to recall bias.

8. For the Title, the reviewer suggests adding “current”. (i.e. Ethnic differences in current smoking....) This has been addressed.

Reviewer #2:

This is interesting research, and it is important to characterize the epidemiology of smoking and smoking cessation. The methods were adequate to answer the scientific questions.

My primary concern is about the interpretation of the results. This is a descriptive paper, with no intervention, and so one needs to be particularly careful about how to use the findings to inform policy. This appears in a number of ways:

1. You found an association between ethnic group status and sociodemographic status. You described this as follows: "SES contributed to the higher prevalence of smoking among Turkish, Surinamese and Moroccan men and Turkish women, but diminished the estimates for Surinamese, Moroccan and Ghanaian women who were less likely to smoke than the Dutch." What do you mean, "SES contributed"? This implies that somehow SES is the driver for smoking behavior independent of ethnic status, and suggests that SES is the cause of smoking. This just does not make sense.

Instead, it appears that various ethnic groups have different rates of smoking and cessation, and these groups also have low SES. But there is no reason to believe that the SES itself was a cause of greater smoking. If you are in fact drawing this conclusion, you need to justify it in more detail.

My own interpretation would be a bit more concrete: certain groups defined by ethnic status and sex have higher and lower rates of smoking, and might benefit from targeted stop-smoking interventions.

Thank you for your observation. Lower SES is a risk factor for smoking, and the ethnic groups studied tended to be of lower SES compared to the Dutch host population. The fact that the differences were less marked (although differences remained) following adjustment for SES strongly suggests that some of the differences observed were attributable to SES differences between the groups. However, this does not alter the overall conclusion, and we agree with the referee that the findings suggest a targeted intervention strategy.

We have adjusted the conclusion as follows. "Although differences were less marked after adjustment, particularly for socioeconomic factors, the findings still suggest that some ethnic minority groups may benefit from targeted cessation interventions."

2. While it is generally known that smoking is unhealthy, there is not necessarily a guaranteed and uniform benefit from cessation at different age groups. Your discussion about "increasing hazard ratios for acute myocardial infarction and stroke across medium and lower SES ethnic groups compared to the higher SES groups" is thus very speculative. The claim that "reducing smoking and or increasing cessation among ethnic minority groups can help reduce the high all-cause mortality and cardiovascular disease incidence already seen among Turkish and Surinamese in the Netherlands" sounds plausible, but is also outside the scope of your research (since you did not examine any effects of smoking). Invoking both SES and ethnic status adds to the confusion. I recommend removing this entire section.

To avoid being speculative, this section has been removed as recommended, see pages 19-20.

3. The home country vs immigrant rates were an interesting result, which deserved more attention and consideration. At the bottom of page 16 (page 18), it was unclear which were the home country rates and which were the Amsterdam rates. I think the home country was the first one. The percentages did not correspond to the prevalence values in Table 2, so I was a bit confused. (And why is this comparison with people in Amsterdam instead of the entire sample?)

This finding deserves more attention, since it points to a potential prevention approach, such as “catching” people at the time that they immigrate. (This would be true if more ethnic minority individuals smoked after immigrating, which I think is the case.) Why do you think, based on your findings, that immigrants smoke at different rates? (Potentially it is mediated by SES, but empirically answering that question would depend on knowing SES before and after immigration.)

The percentages in Table 2 are crude percentages, and the one we used to compare to the home countries were age-standardized prevalences as shown in figure 1, hence the slight difference between the 2. To make clearer, the country names were inserted after each prevalence for the first comparison between women in home country and country of origin, on page 19. We wanted to compare the prevalence of smoking in each ethnic minority group living in Amsterdam (where the HELIUS study was conducted) to recent estimates of smoking for men and women in the original home country, i.e. Suriname, Turkey, Morocco or Ghana.

An explanation for the reasons why smoking differs by ethnic group after adjustment for SES is now included on page 21.

“Although SES contributed to ethnic differences in smoking and cessation, it did not entirely explain the ethnic differences suggesting that other factors may play a role. A possible explanation is that smoking behaviour may be influenced by cultural norms present in the ethnic group, as those sharing a particular ethnic identity tend to share similar attitudes towards smoking.

We have revised other areas to make clearer and to correct any observed mistakes.

VERSION 2 – REVIEW

REVIEWER	Chung-Il Wi Mayo Clinic Rochester United States
REVIEW RETURNED	12-May-2017

GENERAL COMMENTS	All concerns from reviewer were clarified and addressed.
--

REVIEWER	Stephen Thielke University of Washington United States
REVIEW RETURNED	11-May-2017

GENERAL COMMENTS	The authors adequately addressed the concerns about the interpretation of the results. This study contributes to the understanding of smoking behaviors.
--